# Irritable Bowel Syndrome Is an Independent Risk Factor for Developing Opioid Use Disorder in Patients with Inflammatory Bowel Disease

**DOI:** 10.3390/jpm13060917

**Published:** 2023-05-30

**Authors:** Yuhan Fu, Michael Kurin, Marc Landsman, Ronnie Fass, Gengqing Song

**Affiliations:** 1Department of Internal Medicine, Metrohealth Medical Center/Case Western Reserve University, Cleveland, OH 44109, USA; yfu@metrohealth.org; 2Department of Gastroenterology, Metrohealth Medical Center/Case Western Reserve University, Cleveland, OH 44109, USA

**Keywords:** irritable bowel syndrome, opioid-related disorders, ulcerative colitis, Crohn’s disease

## Abstract

Background: Symptoms of IBS can fluctuate even when IBD is in clinical remission. Patients with IBD are at an increased risk of developing opioid addiction. The aim of the study was to determine whether IBS is an independent risk factor for developing opioid addiction and related gastrointestinal symptoms in patients with IBD. Methods: We identified patients with Crohn’s disease (CD)+IBS and ulcerative colitis (UC) + IBS using TriNetX. The control groups consisted of patients with CD or UC alone without IBS. The main outcome was to compare the risks of receiving oral opioids and developing opioid addiction. A subgroup analysis was performed by selecting patients who were prescribed oral opioids and to compare with those not prescribed opioids. Gastrointestinal symptoms and mortality rates were compared in the cohorts. Results: Patients with concomitant IBD and IBS were more likely to be prescribed oral opioids (24.6% vs. 17.2% for CD; 20.2% vs. 12.3% for UC, *p* < 0.0001) and develop opioid dependence or abuse (*p* < 0.05). The subset of patients who were prescribed opioids are more likely to develop gastroesophageal reflux disease, ileus, constipation, nausea, and vomiting (*p* < 0.05). Conclusions: IBS is an independent risk factor for IBD patients to receive opioids and develop opioid addiction.

## 1. Introduction

Inflammatory bowel disease (IBD) consists of Crohn’s disease (CD) and ulcerative colitis (UC), both of which are characterized by chronic inflammation of the gastrointestinal tract [1,2]. CD can affect the entire gastrointestinal tract, whereas UC is restricted to the colon. Abdominal pain is one of the major symptoms of IBD, especially for those with coexisting irritable bowel syndrome (IBS) [3,4]. Chronic abdominal pain could persist even when IBD is in clinical remission, which can be debilitating and significantly affect patients’ quality of life [5,6]. However, patients’ self-reported symptoms may not correlate with disease activity and inflammation [7,8]. Patients with overlapping IBD-IBS are associated with increased psychiatric diagnoses and lower quality of life [9]. It was postulated that IBD and IBS share many different mechanisms, including increased mucosal permeability [10], increased production of bio-mediators [11], abnormal enteric nerves [12], psychological stress [13,14], and gut microbiota dysbiosis [4,15,16].

Acute pain in IBD could be due to active inflammation, fistulas, abscesses, strictures, adhesions, bowel obstruction, and dysmotility [17]. These are treated by addressing the underlying cause. Management of chronic abdominal pain in the absence of active inflammation and IBD-related complications can be challenging [18]. Causes of chronic abdominal pain in the absence of inflammation can include surgical adhesions, fibrostenotic bowel, small intestinal bacterial overgrowth (SIBO), post-surgical complications, or disorders of gut-brain interaction, or any other etiologies like those in the general population (cholecystitis, pancreatitis, etc.). A meta-analysis in 2020 reported that the overall prevalence of IBS in IBD patients was 32.5%; specifically, the prevalence of IBS in patients with CD and UC in remission was estimated to be 36.6% and 28.7%, respectively [19]. 

Even though against recommendations, opioids are commonly prescribed to control chronic abdominal pain in patients with IBD in the real-world clinical setting, which may result in an increased risk of developing opioid addiction in this population [20]. Current guidelines do not support the use of opioids except in acute IBD-related admission. Opioids are well known to cause various gastrointestinal side effects, such as delaying gastrointestinal transit, constipation, nausea, vomiting, bloating, and gastroesophageal reflux disease [21,22]. In addition, opioids are associated with increased healthcare utilization, mortality, and decreased quality of life in patients with IBD [23,24,25]. Patients with coexisting CD and functional gastrointestinal disorders are especially prone to chronic opioid use [26]. Previously, it has been reported that patients with overlapping IBD-IBS were associated with increased opioid use and adverse outcomes in a cross-sectional study [27]. Nevertheless, large-scale database studies addressing this topic are currently lacking. It is essential for clinicians to be aware of the potential risks of opioid misuse in this particular population and refrain from prescribing opioids for the management of functional abdominal pain in IBD patients.

The aim of the study was to assess the risks of opioid prescription use and related complications in patients with overlapping IBD-IBS using a large clinical database. Simultaneously, we aim to raise awareness among clinicians regarding the potential over-prescription of opioids in patients with concurrent IBD and IBS, particularly in light of the ongoing opioid epidemic in the US.

## 2. Materials and Methods

### 2.1. Data Source

TriNetX is a global federated health research network that provided access to aggregated de-identified electronic healthcare record (EHR) data across 93 large healthcare organizations (HCOs) with over 120 million patients. The majority of HCOs are from the US and Europe. This database provides real-time longitudinal patient information, which allows for customizable cohort selection. It contains information including diagnoses, procedures, medications, and laboratory values from both inpatient and outpatient settings. In this study, patients were enrolled from the US Collaborative Network of the TriNetX platform. The US collaborative Network contained 56 HCOs with over 92 million patients by May 2023. Review and approval by Metrohealth Medical Center Institutional Review Board (IRB) was exempted since TriNetX is a de-identified database without the involvement of any identifiable patients’ personal information.

### 2.2. Cohort Definitions

A cohort study was performed using the TriNetX database. We identified four separate study populations: concomitant CD and IBS (CD+IBS), concomitant UC and IBS (UC+IBS), CD+IBS with opioids, and UC+IBS with opioids. There were four control groups: CD, UC, CD+IBS without opioids, and UC+IBS without opioids (Figure 1).

#### 2.2.1. CD/UC+IBS Cohorts

We identified all patients who had a colonoscopy (UMLS:SNOMED:73761001 or UMLS:CPT:1022231) within 1 year before the diagnosis of UC (UMLS:ICD10CM:K51). We identified patients who had completed colonoscopy, or computed tomography of abdomen and pelvis, or Magnetic Resonance abdomen and pelvis, or Computed Tomography/Magnetic Resonance Enterography (CTE or MRE) within 1 year before the diagnosis of CD (UMLS:ICD10CM:K50). Patients needed to be subsequently diagnosed with IBS (UMLS:ICD10CM:K58) within 5 years of the diagnosis of CD or UC. A normal stool calprotectin (UMLS:LNC:38445-3, between 0.00 and 200.00 ug/g), erythrocyte sedimentation rate (ESR, TNX:9066, between 0.00 and 20.00 mm/h) or C-reactive protein (CRP, TNX:9063, between 0.00 and 10.00 mg/L) level within 3 months of the diagnosis of IBS was required to ensure that IBD was in remission when IBS was diagnosed. 

#### 2.2.2. CD/UC Cohorts

Two control groups consisted of patients with CD and UC without IBS. We identified all patients who had a colonoscopy within 1 year before the diagnosis of UC. We identified all patients who had a colonoscopy, or Computed Tomography of the abdomen and pelvis, or Magnetic Resonance abdomen and pelvis, or CTE/MRE within 1 year before the diagnosis of CD. Patients with a diagnosis of IBS were excluded from the cohorts.

#### 2.2.3. CD/UC+IBS with Opioids Cohorts

A subgroup analysis was performed by selecting patients (CD+IBS and UC+IBS) who were prescribed oral opioids (NLM:VA:CN101, oral product). Patients needed to be prescribed oral opioids within 5 years after the diagnosis of CD or UC. 

#### 2.2.4. CD/UC+IBS without Opioids Cohorts

Two control groups consisted of patients (CD+IBS and UC+IBS) who were not prescribed any forms of opioids (including topical, injection, oral, and intranasal) within 5 years after the diagnosis of CD or UC.

### 2.3. Outcome Measures

We compared the risks of 5-year incident prescription of oral opioids (NLM:VA:CN101), oral oxycodone (NLM:RXNORM:7804), oral hydromorphone (NLM:RXNORM:3423), oral morphine (NLM:RXNORM:7052), oral hydrocodone (NLM:RXNORM:5489), oral tramadol (NLM:RXNORM:10689), opioid dependence (UMLS:ICD10CM:F11.2), and opioid abuse (UMLS:ICD10CM:F11.2) in CD (UC)+IBS and CD (UC) cohorts. Opioid use disorder included both opioid dependence and opioid abuse. The index event was defined as the onset of CD or UC in each cohort. We excluded patients with outcomes prior to the diagnosis of CD or UC.

We compared the 5-year incidence of new onset gastroesophageal reflux disease (GERD), (UMLS:ICD10CM:K21), ileus (UMLS:ICD10CM:K56.0 or K56.7), constipation (UMLS:ICD10CM:K59.0), nausea and vomiting (UMLS:ICD10CM:R11) in CD/UC+IBS with and without opioids. We also compared the 5-year incidence of emergency department (ED) visits and inpatient admissions in both cohorts. The index event was defined as the prescription of oral opioids. Patients with outcomes before the time window were excluded.

### 2.4. Statistical Analysis

Chi-square tests and analysis of variance (ANOVA) were used to compare categorical and continuous parameters. We used the TriNetX built-in algorithm to perform the propensity score matching for baseline characteristics, which was based on 1:1 nearest neighbor matching with a caliper of 0.1 SD. We accounted for covariates, including age at the index event, gender, race, and ethnicity. We also accounted for covariates that could contribute to opioid use, including chronic pain syndrome, osteoarthritis, rheumatoid arthritis, low back pain, mental disorders, inflammatory polyarthropathies, radiculopathy, cervicalgia, and thoracic pain. We obtained the number of patients with individual outcomes and the risks of the outcomes in the cohorts. Risk is defined as the fraction of patients with the outcomes within the 5-year window of the index event. Odds ratios (OR) with a 95% confidence interval were obtained to compare the cohorts. *p* < 0.05 was deemed statistically significant. Kaplan–Meier survival analyses and log-rank tests were performed to compare the survival probability in CD/UC+IBS with and without opioids cohorts. The hazard ratio (HR) for the risk of death with a 95% confidence interval was obtained by the Cox regression model.

## 3. Results

### 3.1. Population

There was a total of 8397 patients with concomitant CD and IBS (64.9% female, mean age when diagnosed with CD: 38.5 ± 18.4 years old) and 106,349 patients with CD alone (41.9% female, mean age when diagnosed with CD: 41.9 ± 19.6 years old) (Table 1). There was a total of 5015 patients with concomitant UC and IBS (62.7% female, mean age when diagnosed with UC: 40.2 ± 19.8 years old) and 72,944 patients with UC alone (49.5% female, mean age when diagnosed with UC: 47.4 ± 19.5 years old) (Table 2).

Overall, forty-seven percent of patients with concomitant CD and IBS were prescribed oral opioids at least once after the diagnosis of CD (n = 3955), and fifteen percent of patients were never prescribed any forms of opioids after the diagnosis of CD (n = 1249) before propensity score matching (Table 3). Those who were prescribed oral opioids had an average age of 40.9 years old which was significantly older than those who were not prescribed opioids (*p* < 0.001). Female patients were more likely to be prescribed opioids than males (68.5 vs. 31.5%).

In addition, forty-one percent of patients with concomitant UC and IBS were prescribed oral opioids at least once after UC was diagnosed (n = 2081), and seventeen percent of patients were never prescribed any forms of opioids (n = 834) (Table 4). The average age of those who were prescribed oral opioids was higher than those who were not (43.6 vs. 33.5 years old). More female patients were prescribed opioids than males (65.9 vs. 34.1%).

### 3.2. Opioid Prescription and Opioid Use Disorder

Patients with CD who subsequently developed IBS were more likely to be prescribed oral opioids than patients without IBS after propensity score matching, including all oral opioids (24.6 vs. 17.2%; OR 1.57, *p* < 0.0001), oral oxycodone (18.5 vs. 12.0%; OR 1.66, *p* < 0.0001), oral hydromorphone (5.8 vs. 3.2%; OR 1.88, *p* < 0.0001), oral morphine (4.8 vs. 2.7%; OR 1.81, *p* < 0.0001), oral hydrocodone (13.0 vs. 8.4%; OR 1.62, *p* < 0.0001), and oral tramadol (11.4 vs. 6.4%, OR 1.88, *p* < 0.0001) (Table 5). Patients with CD who developed IBS were more likely to develop opioid use disorder than those without IBS, including opioid dependence (3.8 vs. 1.6%; OR 2.46, *p* < 0.0001) and opioid abuse (1.8 vs. 0.8%; OR 2.30, *p* < 0.0001).

Similarly, patients with UC who developed IBS were at a higher risk to be prescribed various oral opioids than those without IBS after propensity score matching, including all oral opioids (20.2 vs. 12.3%; OR 1.80, *p* < 0.0001), oral oxycodone (15.8 vs. 8.5%; OR 2.03, *p* < 0.0001), oral hydromorphone (4.4 vs. 1.6%; OR 2.81, *p* < 0.0001), oral morphine (3.6 vs. 1.8%; OR 2.06, *p* < 0.0001), oral hydrocodone (10.3 vs. 5.6%; OR 1.94, *p* < 0.0001), and oral tramadol (9.0 vs. 4.4%, OR 2.17, *p* < 0.0001) (Table 6). Patients with UC who developed IBS were more likely to develop opioid use disorder, including opioid dependence (2.2 vs. 0.8%; OR 2.60, *p* < 0.0001) and opioid abuse (0.8 vs. 0.4%; OR 2.09, *p* = 0.016) than those without IBS.

### 3.3. Opioids-Related Gastrointestinal Complications and Mortality

After propensity score matching was performed, patients with concomitant CD and IBS who were prescribed oral opioids were more likely to develop gastrointestinal symptoms and complications than patients who were not prescribed opioids: including GERD (35.7 vs. 22.9%; OR 1.87, *p* < 0.0001), ileus (8.6 vs. 0.9%; OR 10.29, *p* < 0.0001), constipation (27.2 vs. 16.0%; OR 1.96, *p* < 0.0001), nausea and vomiting (48.7 vs. 21.1%; OR 3.54, *p* < 0.0001) (Table 7). Patients who were prescribed oral opioids were also more likely to have ED visits (56.8 vs. 24.4%; OR 4.07, *p* < 0.0001) and hospital admissions (59.0 vs. 15.2%; OR 8.06, *p* < 0.0001). Furthermore, Kaplan–Meier analysis showed that patients with CD and IBS who were prescribed opioids had poorer survival probabilities within a 5-year follow-up period in terms of all-cause mortality than patients who were not prescribed opioids (survival probability 96.1 vs. 99.0%; HR 3.91; 95% CI 1.73–8.84; log-rank test: *p* = 0.0004).

Similar trends were observed with patients with coexisting UC and IBS. After propensity score matching was performed, patients with concomitant UC and IBS who were prescribed oral opioids were more likely to have gastrointestinal symptoms and complications than patients who were not prescribed opioids: including GERD (28.7 vs. 19.8%; OR 1.63, *p* = 0.0010), Ileus (8.5 vs. 0%; *p* < 0.0001), constipation (25.0 vs. 15.7%; OR 1.79, *p* = 0.0001), nausea and vomiting (40.5 vs. 15.9%; OR 3.61, *p* < 0.0001) (Table 8). Those who were prescribed oral opioids were more likely to have ED visits (53.4 vs. 20.8%; OR 4.38, *p* < 0.0001) and inpatient admissions (58.2 vs. 17.0%; OR 6.81, *p* < 0.0001). Kaplan–Meier analysis showed that patients with CD and IBS who were prescribed opioids had poor survival probabilities within a 5-year follow-up period in terms of all-cause mortality than patients who were not prescribed opioids (survival probability 96.8 vs. 99.0%; HR 3.16; 95% CI 1.17–8.52; log-rank test: *p* = 0.017).

## 4. Discussion

Our study reveals that IBS is an independent risk factor for receiving oral opioids and developing opioid use disorder in patients with IBD. We also found that opioids increased the risks of developing several gastrointestinal complications and resulting in a higher mortality rate and healthcare utilization in patients with concomitant IBD and IBS. This is one of the first studies to determine the risks of opioid prescription and related gastrointestinal complications in concomitant IBD and IBS using a large database.

Our study shows that IBS brings a 1.57 and 1.80-fold increase in the risks of receiving oral opioid analgesics in patients with concomitant CD or UC, respectively. A retrospective study with a sample size of 931 patients showed that chronic prescription opioid use was more common in patients with concomitant CD and functional gastrointestinal disorders (FGID) than in patients with CD alone (44 vs. 18%, *p* < 0.001) [26]. IBS is also a strong risk factor for inpatient narcotic use for IBD patients who were admitted to the hospital [28]. A cross-sectional study showed that patients with concurrent CD-IBS and UC-IBS were associated with higher risks of opioid use (16.5 vs. 10.5%; 9.3 vs. 4.7%) [27]. Our results show a consistent trend with the data reported in the literature: 24.6% of patients with CD and IBS, and 20.2% of patients with UC and IBS were prescribed oral opioid analgesics at least once (*p* < 0.001). Our data also showed a consistent trend with the literature that patients with CD were more likely to suffer from pain than UC [29]. However, our study did not differentiate whether oral opioids were prescribed for short-term use or for chronic use and whether they were prescribed in the inpatient setting or outpatient setting, which may have overestimated the overall risks of opioid prescription.

IBS is a strong risk factor for patients to develop opioid use disorder. We observed a 2.30-fold and 1.92-fold increase in opioid abuse in patients with concomitant CD+IBS and UC+IBS, respectively. It has been reported that IBD is an independent risk factor for a patient to become a heavy opioid user [20]. Moreover, a study has shown that FGID was associated with a 4.5-fold increase in opioid dependence and a 5-fold increase in opioid misuse in patients with UC [30]. Another study reported that opioid misuse in patients with CD is strongly associated with a concurrent diagnosis of FGID [26]. However, the previous studies did not differentiate IBS from other FGIDs. Overall, our results showed a consistent trend with the literature that IBS appears to be a risk factor for opioid-related disorders, and this is one of the first studies to separate IBS from other FGIDs. 

Opioid analgesics are associated with increased risks of developing various gastrointestinal complications in patients with overlapping IBD and IBS, including nausea, vomiting, constipation, GERD, and ileus. The association was the strongest with ileus. Opioids are known to cause various gastrointestinal symptoms, including constipation, nausea, vomiting, and motility issues such as esophageal dysmotility and ileus [31,32,33]. Opioids can also cause narcotic bowel syndrome, characterized by the development or worsening of abdominal pain with continuous or escalating doses of narcotics [34]. However, this is the first study to investigate the gastrointestinal side effects of opioids in patients with concurrent IBD and IBS. It is reasonable to postulate that opioids may be contributing to the worsening gastrointestinal symptoms and side effects.

Patients with concurrent IBD and IBS who were prescribed oral opioids appear to be high utilizers of the healthcare system. They were more likely to have ED visits and be admitted to the hospital, though we could not determine the chief complaints leading to the specific admission or ED visit. It was previously reported that IBS symptoms affected more than 60% of patients with IBD during a 6-year follow-up period and they tend to be high utilizers of the healthcare system [35]. This population warrants more frequent outpatient monitoring by gastroenterologists and their primary care physicians to avoid the overutilization of the healthcare system. 

Furthermore, our study shows that opioid analgesics are associated with a higher all-cause mortality rate in patients with concomitant IBD and IBS. Previous studies have shown that opioid analgesics are associated with a higher mortality rate in patients with IBD [20,36]. We demonstrated consistent results for patients with overlapping IBD and IBS. 

We would like to emphasize that the use of opioid prescription medications is associated with worse clinical outcomes and higher healthcare utilization. Even though prescribing opioids may be an easy decision for clinicians to make to satisfy patients’ immediate needs and avoid lengthy discussions, it may easily feed into a vicious cycle and even contribute to worsening mortality in the long term. Clinicians should prioritize the principle of "do no harm" and avoid prescribing opioids when managing functional abdominal pain in patients with IBD. Future research areas should be focused on better symptom control for IBD patients with overlapping IBS features to avoid opioid analgesic use. Current guideline recommends a low FODMAP diet and psychological therapies in managing functional GI symptoms in IBD patients [37]. A low FODMAP diet has been shown to reduce IBS-like symptoms and improve quality of life in IBD patients [38,39]. Clinicians should offer such guideline-compliant therapy options to patients instead of opioids.

There are several limitations to the current study. The data obtained were aggregated, and it was not possible to access patients’ data on an individual level. Certain confounding factors may not have been accounted for. The diagnoses are based on billing codes and could not be verified. For example, it is difficult to tell if IBS and IBD were diagnosed by gastroenterologists or not and whether the diagnoses were accurate or not. ESR and CRP can be nonspecific, thus a normal value can still be associated with active IBD; whereas elevated fecal calprotectin is more specific for assessing disease activity.

In conclusion, this is one of the first studies to evaluate the risks of opioid prescription and related complications in patients with concomitant IBD and IBS. We found that IBS is an independent risk factor for developing opioid addiction in patients with IBD and that patients who were prescribed opioids are more likely to have gastrointestinal side effects and higher mortality than those with IBD alone. Clinicians should be aware of the risks of opioid-related complications in this population and refrain from prescribing opioids in managing IBS symptoms in IBD.

## Figures and Tables

**Figure 1 jpm-13-00917-f001:**
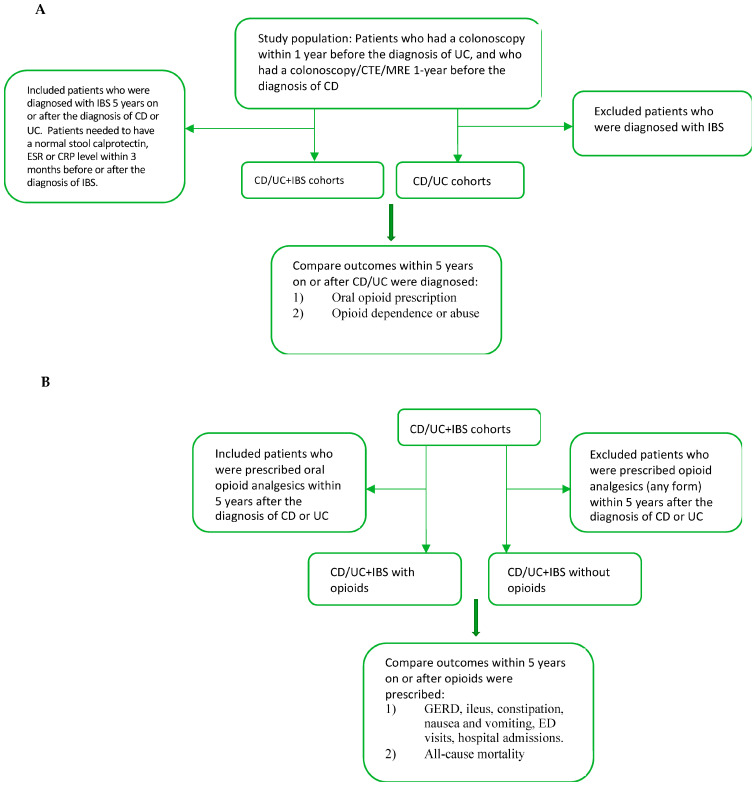
Study design flow chart. (**A**): Cohort selection of CD/UC with and without IBS. (**B**): Subgroup analyses of CD/UC+IBS with and without opioids. Abbreviations: CTE: Computed Tomography Enterography, MRE: Magnetic Resonance Enterography, ESR:Erythrocyte sedimentation rate, CRP: C-reactive Protein, ED: Emergency department.

**Table 1 jpm-13-00917-t001:** Baseline characteristics in patients with concomitant Crohn’s disease and IBS.

		Unadjusted Data				After Propensity-Score Matching			
		CD+IBS	CD without IBS	*p*-Value	SD	CD+IBS	CD without IBS	*p*-Value	SD
N of patients		8397	106349			8246	8246		
Age		38.5 ± 18.4	41.9 ± 19.6	<0.001	0.180	38.5 ± 18.4	38.5 ± 18.6	0.984	<0.001
Gender	Female	5357 (64.9%)	54,657 (52.2%)	<0.001	0.259	5347 (64.8%)	5322 (64.5%)	0.684	0.006
	Male	2899 (35.1%)	49,975 (47.8%)	<0.001	0.259	2899 (35.2%)	2923 (35.4%)	0.696	0.006
Race	White	6643 (80.5%)	79,655 (76.1%)	<0.001	0.105	6634 (80.5%)	6719 (81.5%)	0.092	0.026
	Unknown	743 (9.0%)	11837 (11.3%)	<0.001	0.077	742 (9.0%)	682 (8.3%)	0.096	0.026
	Black or African American	740 (9.0%)	11045 (10.6%)	<0.001	0.054	740 (9.0%)	729 (8.8%)	0.764	0.005
	Asian	93 (1.1%)	1769 (1.7%)	<0.001	0.048	93 (1.1%)	93 (1.1%)	1.000	<0.001
Ethnicity	Hispanic or Latino	404 (4.9%)	5583 (5.3%)	0.085	0.020	404 (4.9%)	384 (4.7%)	0.465	0.011
	Not Hispanic or Latino	6902 (83.6%)	77,536 (74.1%)	<0.001	0.234	6892 (83.6%)	6982 (84.7%)	0.055	0.03
	Unknown	950 (11.5%)	21,523 (20.6%)	<0.001	0.249	950 (11.5%)	880 (10.7%)	0.083	0.027
Diagnosis	Chronic pain syndrome	288 (3.5%)	1508 (1.4%)	<0.001	0.132	279 (3.4%)	262 (3.2%)	0.457	0.012
	Osteoarthritis	1372 (16.6%)	11,745 (11.2%)	<0.001	0.156	1363 (16.5%)	1326 (16.1%)	0.435	0.012
	Rheumatoid arthritis	461 (5.6%)	2611 (2.5%)	<0.001	0.157	454 (5.5%)	440 (5.3%)	0.630	0.007
	Low back pain	1490 (18.0%)	10,929 (10.4%)	<0.001	0.219	1480 (17.9%)	1458 (17.7%)	0.654	0.007
	Mental disorder	4078 (49.4%)	36,347 (34.7%)	<0.001	0.300	4068 (49.3%)	4085 (49.5%)	0.791	0.004
	Inflammatory polyarthropathies	1154 (14.0%)	8284 (7.9%)	<0.001	0.195	1144 (13.9%)	1113 (13.5%)	0.482	0.011
	Radiculopathy	676 (8.2%)	4694 (4.5%)	<0.001	0.152	667 (8.1%)	649 (7.9%)	0.605	0.008
	Cervicalgia	1211 (14.7%)	8977 (8.6%)	<0.001	0.191	1201 (14.6%)	1152 (14.0%)	0.275	0.017
	Pain in thoracic spine	356 (4.3%)	2210 (2.1%)	<0.001	0.125	350 (4.2%)	321 (3.9%)	0.253	0.018

**Table 2 jpm-13-00917-t002:** Baseline characteristics in patients with concomitant ulcerative colitis and IBS.

		Unadjusted Data				After Propensity-Score Matching			
		UC+IBS	UC without IBS	*p*-Value	SD	UC+IBS	UC without IBS	*p*-Value	SD
N of patients		5015	72,944			4911	4911		
Age		40.2 ± 19.8	47.4 ± 19.5	<0.001	0.369	40.2 ± 19.8	40.0 ± 19.7	0.671	0.009
Gender	Female	3082 (62.7%)	35,270 (49.5%)	<0.001	0.269	3075 (62.6%)	3056 (62.2%)	0.692	0.008
	Male	1836 (37.3%)	36,037 (50.5%)	<0.001	0.268	1836 (37.4%)	1853 (37.7%)	0.723	0.007
Race	White	3916 (79.6%)	53,787 (75.4%)	<0.001	0.101	3910 (79.6%)	3957 (80.6%)	0.235	0.024
	Unknown	512 (10.4%)	8801 (12.3%)	<0.001	0.061	511 (10.4%)	474 (9.7%)	0.214	0.025
	Black or African American	383 (7.8%)	6855 (9.6%)	<0.001	0.065	383 (7.8%)	382 (7.8%)	0.970	0.001
	Asian	83 (1.7%)	1621 (2.3%)	0.007	0.042	83 (1.7%)	73 (1.5%)	0.420	0.016
Ethnicity	Hispanic or Latino	292 (5.9%)	4265 (6.0%)	0.902	0.002	292 (5.9%)	300 (6.1%)	0.734	0.007
	Not Hispanic or Latino	4039 (82.1%)	52478 (73.6%)	<0.001	0.207	4032 (82.1%)	4086 (83.2%)	0.150	0.029
	Unknown	587 (11.9%)	14573 (20.4%)	<0.001	0.232	587 (12.0%)	525 (10.7%)	0.048	0.040
Diagnosis	Chronic pain syndrome	141 (2.9%)	927 (1.3%)	<0.001	0.110	137 (2.8%)	103 (2.1%)	0.026	0.045
	Osteoarthritis	945 (19.2%)	11,958 (16.8%)	<0.001	0.064	938 (19.1%)	893 (18.2%)	0.244	0.024
	Rheumatoid arthritis	282 (5.7%)	1928 (2.7%)	<0.001	0.151	275 (5.6%)	270 (5.5%)	0.826	0.004
	Low back pain	928 (18.9%)	9407 (13.2%)	<0.001	0.155	922 (18.8%)	863 (17.6%)	0.123	0.031
	Mental disorder	2435 (49.5%)	25,050 (35.1%)	<0.001	0.294	2428 (49.4%)	2450 (49.9%)	0.657	0.009
	Inflammatory polyarthropathies	726 (14.8%)	7291 (10.2%)	<0.001	0.138	719 (14.6%)	665 (13.5%)	0.117	0.032
	Radiculopathy	435 (8.8%)	4552 (6.4%)	<0.001	0.093	432 (8.8%)	373 (7.6%)	0.030	0.044
	Cervicalgia	757 (15.4%)	6549 (9.2%)	<0.001	0.190	750 (15.3%)	717 (14.6%)	0.350	0.019
	Pain in thoracic spine	195 (4.0%)	1827 (2.6%)	<0.001	0.079	194 (4.0%)	163 (3.3%)	0.095	0.034

**Table 3 jpm-13-00917-t003:** Baseline characteristics of concomitant Crohn’s disease and IBS with and without opioid analgesics.

		Unadjusted Data				After Propensity-Score Matching			
		CD+IBS							
		With Opioids	Without Opioids	*p*-Value	SD	With Opioids	Without Opioids	*p*-Value	SD
N of patients		3955	1249			1115	1115		
Age		40.9 ± 17.4	31.9 ± 19.3	<0.001	0.488	33.9 ± 16.9	33.9 ± 19.2	0.984	0.001
Gender	Female	2674 (68.5%)	718 (58.2%)	<0.001	0.214	672 (60.3%)	696 (62.4%)	0.297	0.044
	Male	1231 (31.5%)	515 (41.8%)	<0.001	0.214	443 (39.7%)	419 (37.6%)	0.297	0.044
Race	White	3247 (83.1%)	993 (80.5%)	0.035	0.068	924 (82.9%)	910 (81.6%)	0.438	0.033
	Unknown	283 (7.2%)	131 (10.6%)	<0.001	0.119	107 (9.6%)	106 (9.5%)	0.943	0.003
	Black or African American	332 (8.5%)	77 (6.2%)	0.011	0.086	65 (5.8%)	77 (6.9%)	0.298	0.044
	Asian	24 (0.6%)	27 (2.2%)	<0.001	0.134	15 (1.3%)	17 (1.5%)	0.722	0.015
Ethnicity	Hispanic or Latino	221 (5.7%)	76 (6.2%)	0.508	0.021	73 (6.5%)	70 (6.3%)	0.795	0.011
	Not Hispanic or Latino	3287 (84.2%)	989 (80.2%)	0.001	0.104	906 (81.3%)	907 (81.3%)	0.957	0.002
	Unknown	397 (10.2%)	168 (13.6%)	0.001	0.107	136 (12.2%)	138 (12.4%)	0.897	0.005
Diagnosis	Chronic pain syndrome	191 (4.9%)	13 (1.1%)	<0.001	0.227	14 (1.3%)	13 (1.2%)	0.846	0.008
	Osteoarthritis	932 (23.9%)	94 (7.6%)	<0.001	0.457	100 (9.0%)	93 (8.3%)	0.598	0.022
	Rheumatoid arthritis	302 (7.7%)	27 (2.2%)	<0.001	0.257	32 (2.9%)	27 (2.4%)	0.509	0.028
	Low back pain	971 (24.9%)	103 (8.4%)	<0.001	0.455	101 (9.1%)	102 (9.1%)	0.941	0.003
	Mental disorder	2497 (63.9%)	362 (29.4%)	<0.001	0.739	356 (31.9%)	362 (32.5%)	0.786	0.012
	Inflammatory polyarthropathies	745 (19.1%)	69 (5.6%)	<0.001	0.419	87 (7.8%)	69 (6.2%)	0.135	0.063
	Radiculopathy	433 (11.1%)	44 (3.6%)	<0.001	0.292	40 (3.6%)	43 (3.9%)	0.737	0.014
	Cervicalgia	815 (20.9%)	102 (8.3%)	<0.001	0.363	105 (9.4%)	102 (9.1%)	0.827	0.009
	Pain in thoracic spine	234 (6.0%)	34 (2.8%)	<0.001	0.159	27 (2.4%)	33 (3.0%)	0.432	0.033

**Table 4 jpm-13-00917-t004:** Baseline characteristics of concomitant ulcerative colitis and IBS with and without opi-oid analgesics.

		UC+IBS							
		With Opioids	Without Opioids	*p*-Value	SD	With Opioids	Without Opioids	*p*-Value	SD
N of patients		2081	834			713	713		
Age		43.6 ± 18.5	33.5 ± 21.2	<0.001	0.507	36.6 ± 17.7	36.7 ± 20.7	0.940	0.004
Gender	Female	1355 (65.9%)	467 (57.2%)	<0.001	0.179	431 (60.4%)	434 (60.9%)	0.871	0.009
	Male	701 (34.1%)	349 (42.8%)	<0.001	0.179	282 (39.6%)	279 (39.1%)	0.871	0.009
Race	White	1676 (81.5%)	650 (79.7%)	0.252	0.047	602 (84.4%)	569 (79.8%)	0.023	0.121
	Unknown	205 (10.0%)	98 (12.0%)	0.109	0.065	66 (9.3%)	86 (12.1%)	0.086	0.091
	Black or African American	141 (6.9%)	47 (5.8%)	0.283	0.045	27 (3.8%)	47 (6.6%)	0.017	0.127
	Asian	24 (1.2%)	18 (2.2%)	0.037	0.081	14 (2.0%)	10 (1.4%)	0.410	0.044
Ethnicity	Hispanic or Latino	133 (6.5%)	59 (7.2%)	0.461	0.030	46 (6.5%)	59 (8.3%)	0.187	0.070
	Not Hispanic or Latino	1668 (81.1%)	650 (79.7%)	0.367	0.037	575 (80.6%)	556 (78.0%)	0.214	0.066
	Unknown	255 (12.4%)	107 (13.1%)	0.605	0.021	92 (12.9%)	98 (13.7%)	0.640	0.025
Diagnosis	Chronic pain syndrome	74 (3.6%)	10 (1.2%)	0.001	0.155	10 (1.4%)	10 (1.4%)	1.000	<0.001
	Osteoarthritis	573 (27.9%)	74 (9.1%)	<0.001	0.499	74 (10.4%)	73 (10.2%)	0.931	0.005
	Rheumatoid arthritis	152 (7.4%)	17 (2.1%)	<0.001	0.252	23 (3.2%)	17 (2.4%)	0.336	0.051
	Low back pain	512 (24.9%)	61 (7.5%)	<0.001	0.487	69 (9.7%)	61 (8.6%)	0.462	0.039
	Mental disorder	1288 (62.6%)	238 (29.2%)	<0.001	0.713	236 (33.1%)	238 (33.4%)	0.910	0.006
	Inflammatory polyarthropathies	413 (20.1%)	46 (5.6%)	<0.001	0.442	54 (7.6%)	46 (6.5%)	0.407	0.044
	Radiculopathy	243 (11.8%)	33 (4.0%)	<0.001	0.291	46 (6.5%)	33 (4.6%)	0.132	0.080
	Cervicalgia	426 (20.7%)	62 (7.6%)	<0.001	0.383	74 (10.4%)	62 (8.7%)	0.279	0.057
	Pain in thoracic spine	103 (5.0%)	17 (2.1%)	<0.001	0.159	17 (2.4%)	17 (2.4%)	1.000	<0.001

**Table 5 jpm-13-00917-t005:** Opioid prescriptions in patients with concomitant Crohn’s disease and IBS.

		CD+IBS		CD without IBS		CD+IBS vs. CD without IBS		
		N	Risk	N	Risk	OR	95% CI	*p*
Medications	Opioid analgesics	1335	24.6%	938	17.2%	1.57	(1.43, 1.73)	<0.0001
	Oxycodone	1231	18.5%	808	12.0%	1.66	(1.51, 1.83)	<0.0001
	Hydromorphone	457	5.8%	252	3.2%	1.88	(1.61, 2.20)	<0.0001
	Morphine	376	4.8%	212	2.7%	1.81	(1.52, 2.15)	<0.0001
	Hydrocodone	875	13.0%	570	8.4%	1.62	(1.45, 1.82)	<0.0001
	Tramadol	841	11.4%	477	6.4%	1.88	(1.67, 2.11)	<0.0001
Opioid related diagnosis	Opioid dependence	305	3.8%	127	1.6%	2.46	(1.99, 3.03)	<0.0001
	Opioid abuse	146	1.8%	64	0.8%	2.30	(1.72, 3.10)	<0.0001

**Table 6 jpm-13-00917-t006:** Opioid prescriptions in patients with concomitant ulcerative colitis and IBS.

		UC+IBS		UC without IBS		UC+IBS vs. UC without IBS		
		N	Risk	N	Risk	OR	95% CI	*p*
Medications	Opioid analgesics	674	20.2%	437	12.3%	1.80	(1.58, 2.06)	<0.0001
	Oxycodone	633	15.8%	355	8.5%	2.03	(1.77, 2.33)	<0.0001
	Hydromorphone	207	4.4%	77	1.6%	2.81	(2.16, 3.66)	<0.0001
	Morphine	170	3.6%	85	1.8%	2.06	(1.58, 2.68)	<0.0001
	Hydrocodone	424	10.3%	234	5.6%	1.94	(1.65, 2.29)	<0.0001
	Tramadol	394	9.0%	199	4.4%	2.17	(1.82, 2.58)	<0.0001
Opioid related diagnosis	Opioid dependence	104	2.2%	41	0.8%	2.60	(1.80, 3.73)	<0.0001
	Opioid abuse	38	0.8%	20	0.4%	1.92	(1.12, 3.31)	0.016

**Table 7 jpm-13-00917-t007:** Opioid-related complications in concomitant Crohn’s disease and IBS.

	CD+IBS						
	With Opioids		Without Opioids		With Opioids vs. without Opioids		
	N	Risk	N	Risk	OR	95% CI	*p*
GERD	266	35.7%	188	22.9%	1.87	(1.50, 2.33)	<0.0001
Ileus	92	8.6%	10	0.9%	10.29	(5.33, 19.87)	<0.0001
Constipation	225	27.2%	130	16.0%	1.96	(1.54, 2.50)	<0.0001
Nausea and vomiting	295	48.7%	160	21.1%	3.54	(2.80, 4.48)	<0.0001
ED visits	633	56.8%	272	24.4%	4.07	(3.40, 4.88)	<0.0001
Hospital admissions	658	59.0%	169	15.2%	8.06	(6.58, 9.87)	<0.0001

**Table 8 jpm-13-00917-t008:** Opioid-related complications in concomitant ulcerative colitis and IBS.

	UC+IBS						
	With Opioids		Without Opioids		With Opioids vs. without Opioids		
	N	Risk	N	Risk	OR	95% CI	*p*
GERD	135	28.7%	105	19.8%	1.63	(1.22, 2.18)	0.0010
Ileus	58	8.5%	0	0.0%	--	--	<0.0001
Constipation	137	25.0%	88	15.7%	1.79	(1.33, 2.42)	0.0001
Nausea and vomiting	177	40.5%	86	15.9%	3.61	(2.68, 4.87)	<0.0001
ED visits	381	53.4%	148	20.8%	4.38	(3.47, 5.53)	<0.0001
Hospital admissions	415	58.2%	121	17.0%	6.81	(5.33, 8.71)	<0.0001

## Data Availability

Data will be available by contacting the corresponding author upon requests.

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
