# Peer review of "Irritable Bowel Syndrome Is an Independent Risk Factor for Developing Opioid Use Disorder in Patients with Inflammatory Bowel Disease"

_jpm, 2023, doi:10.3390/jpm13060917_

Round 1

Reviewer 1 Report

The paper by Fu et al reports on the effect of a IBS co-diagnosis on opioid use in IBD patients. They find that an IBS co-diagnosis is associated with higher risk for opioid prescriptions and or abuse. Patients on opioids had more GI disease other than IBD. The study design has considerate limitations.

I have the following points to make:

1.       Risk of opioid prescription and or abuse varies a lot in different nations. The US has particularly high rates. It is vitally important to determine what proportion of patients stems from US cohorts.  The findings need to be seen in this light and extrapolation beyond the US may not be applicable.

2.       I think you are mistaken to assume a diagnosis of IBS prior or after a formal IBD diagnosis is the same as Rome criteria identifying IBS symptoms in patients with IBD like in the paper by Ford for example. There are significant issues with your approach:

a.       Patients mislabelled as IBS first with subsequent IBD diagnosis may not have IBS symptoms when IBD is now being treated, yet your search would classify them as IBD+IBS.

b.       While your approach to use FCP is laudable, CRP or ESR are not sufficient to rule out GI inflammation.

c.       By using colonoscopy as the sole diagnostic criterium you likely miss patients with small bowel CD, who could have significant functional symptoms.

d.       In reality few clinicians give a label of IBS or code for this in patients with IBD who have additional functional symptoms.

3.       Opioid exposure yes/no is very crude when applied to 5 year period as someone having a single course of codeine for a broken foot will be classed as exposed the same way you class a patient using morphine daily for 5 years.

4.       How accurate are the diagnoses of opioid dependence and abuse in the cohort? How were these diagnoses reached?

5.       Why did you not examine for complications of IBD such as IBD related hospital admission, surgery, etc?

6.       Why did you not look for comorbidities that ay have led to opioids prescriptions especially musculoskeletal disorders.

7.       You find that 85% of CD/IBS patients were on opioids that seems likely a very large number not recognised in routine clinical practice. I’m not sure how representative your cohort is.

8.       I’m not sure your mortality analysis is solid enough if you don’t account for other factors that may lead to opioid prescriptions.

Minor points:

9.       Abstract: Please avoid the use of ‘complications’ as this suggest complications of IBD like bowel obstruction or fistulas. You simple mention symptoms.

10.   Introduction. Please highlight that while opioids are often prescribed for IBD. Guidelines do not recommend their use outside acute admissions.

11.    

Author Response

  1. Risk of opioid prescription and or abuse varies a lot in different nations. The US has particularly high rates. It is vitally important to determine what proportion of patients stems from US cohorts. The findings need to be seen in this light and extrapolation beyond the US may not be applicable.

We re-ran the study on US collaborative network, which consists of only patients from the US. We were able to demonstrate consistent results with our previous results from global collaboarve network.

  1. I think you are mistaken to assume a diagnosis of IBS prior or after a formal IBD diagnosis is the same as Rome criteria identifying IBS symptoms in patients with IBD like in the paper by Ford for example. There are significant issues with your approach:

In this database study, all the diagnoses are dependent on correct ICD codes/diagnostic coding input. Unfortunately, we were unable to verify the IBS diagnosis by ourselves using the Rome criteria since all data were deidentified and aggregated. We selected patients who were diagnosed with IBS when the IBD is in remission in order to be more accurate. This is the best we could achieve within the limitation of this database platform.

a. Patients mislabeled as IBS first with subsequent IBD diagnosis may not have IBS symptoms when IBD is now being treated, yet your search would classify them as IBD+IBS.

We only selected patients whose IBS diagnosis was made during IBD remission.

b. While your approach to use FCP is laudable, CRP or ESR are not sufficient to rule out GI inflammation.

We understand stool calprotectin, CRP, and ESR may not be sufficient or reliable, however we need to rely on these biomarkers to ensure that IBD is in remission when the IBS diagnosis was made since we don’t have a better approach to ensure that IBD is in remission otherwise.

c. By using colonoscopy as the sole diagnostic criterium you likely miss patients with small bowel CD, who could have significant functional symptoms.

We added CT and MR enterography to the study as well in the revision to capture small bowel CD

d. In reality few clinicians give a label of IBS or code for this in patients with IBD who have additional functional symptoms.

We only selected patients whose IBS diagnosis was made during IBD remission.

  1. Opioid exposure yes/no is very crude when applied to 5-year period as someone having a single course of codeine for a broken foot will be classed as exposed the same way you class a patient using morphine daily for 5 years.

In TriNetX, we couldn’t differentia between short term vs long term opioid prescription unfortunately. We could not tell whether the patient is prescribed opioids for 5-day course vs daily opioid use.

  1. How accurate are the diagnoses of opioid dependence and abuse in the cohort? How were these diagnoses reached?

Opioid dependence and opioid abuse were determined based on ICD diagnostic coding input from the electronic medical records. We don’t have a better way to verify these diagnoses ourselves since all data were aggregated and deidentified.

  1. Why did you not examine for complications of IBD such as IBD related hospital admission, surgery, etc?

We examined ED visits and hospital stays as a potential outcome in the revision. We didn’t add IBD related surgeries and didn’t feel that would fit in the scope of the paper.

  1. Why did you not look for comorbidities that may have led to opioids prescriptions especially musculoskeletal disorders.

We added and accounted for additional confounders in the revision, including osteoarthritis, rheumatoid arthritis, chronic pain, low back pain, inflammatory arthritis, radiculopathy, cervicalgia, thoracic pain and mental/psych disorders

  1. You find that 85% of CD/IBS patients were on opioids that seems likely a very large number not recognized in routine clinical practice. I’m not sure how representative your cohort is.

A significant percentage of patients were on Opioids because we couldn’t differentiate between one time vs long-term opioid prescription.

  1. I’m not sure your mortality analysis is solid enough if you don’t account for other factors that may lead to opioid prescriptions.

The mortality analysis was generated using the TrinetX built-in statistical software which has been verified and proven to be reliable.

Ref: https://support.trinetx.com/hc/en-us/articles/360004087273-How-does-TriNetX-verify-software-that-generates-analytic-results-

Minor points:

9. Abstract: Please avoid the use of ‘complications’ as this suggest complications of IBD like bowel obstruction or fistulas. You simple mention symptoms.

It was addressed in the updated manuscript.

10. Please highlight that while opioids are often prescribed for IBD. Guidelines do not recommend their use outside acute admissions.

It was addressed in the updated manuscript.

Reviewer 2 Report

Crohn’s disease (CD) and most of what resides under the label of ulcerative colitis are two distinct diseases whose causation of pain significantly diverges. The pain caused by strictures, impending bowel inflammation differs from colonic inflammatory neuritis not only in mechanism, but in location and potential intensity.

 The presumed diagnosis of irritable bowel syndrome (for which the is no definitive diagnostic test) is open to question. CD afflicted individuals were more likely to have been prescribed opiates based upon the nature of the pain. Opioids decrease bowel motility and afford pain relief. Knowledge of these factors stands to color their use.

There is the problem as to what irritable bowel syndrome represents. The overlap between IBD and IRS is extensive. The diagnosis of IBS relies on the absence of selective criteria. For CD, the authors are recommended to read. Scanu A. M. et al J. Clin. Microbiol. 2007, p.3883-3890,

There is no place for the use of opiates in CD. The adverse impact of opiates on long-term outcomes is already well documented both by the authors and the literature. A retrospective added affirmation of misuse of these compounds does not need added airing. To do so could suggest that opiates have a valid role in the management of CD, UC (alias IBD).

What was well done were the exclusion criteria utilized to presume inactive disease and the degree of number crunching to achieve what common sense would have dictated and the literature has documented.

Round 2

Reviewer 1 Report

Many thanks for addressing all the points and re-running the analyses.

Author Response

Thank you very much.

Reviewer 2 Report

The use addictive medication for chronic pain is borderline malpractice. Plication of symptomology without addressing the various underlying mechanisms of causation contributes to well documented increased morbidity and probable premature mortality associated with opiate therapy for chronic disorders of the gastrointestinal tract. It is central in medicine that in, attempting to do good, one does not do evil.

What is IBS is underplayed. The contention that IDS is a forme fruste of CD is understated.

The manuscript reiterates previous knowledge. By repetition, it indirectly infers that opiate therapy is a tool in the pain management which plays  against the reason for doing the study.

this is a discussion that must be incorporated into the discussion and introduction portions of the paper.

For a technical point of view , the study is reasonable

Author Response

We appreciate the valuable suggestions. In response, we have made significant improvements to the paper by incorporating a more comprehensive discussion regarding the potential harms associated with prescribing opioids. Specifically, we have highlighted and emphasized our strong recommendation against the use of opioids in managing chronic functional abdominal pain among patients with IBD. The expanded sections in the introduction and discussion now provide a more detailed analysis of this issue, further underscoring the importance of alternative therapeutic approaches.